# FORGE: A Novel Scoring System to Predict the MIB-1 Labeling Index in Intracranial Meningiomas

**DOI:** 10.3390/cancers13143643

**Published:** 2021-07-20

**Authors:** Johannes Wach, Tim Lampmann, Ági Güresir, Patrick Schuss, Hartmut Vatter, Ulrich Herrlinger, Albert Becker, Michael Hölzel, Marieta Toma, Erdem Güresir

**Affiliations:** 1Department of Neurosurgery, University Hospital Bonn, 53127 Bonn, Germany; tim.lampmann@ukbonn.de (T.L.); agi.gueresir@ukbonn.de (Á.G.); patrick.schuss@ukbonn.de (P.S.); hartmut.vatter@ukbonn.de (H.V.); erdem.gueresir@ukbonn.de (E.G.); 2Division of Clinical Neurooncology, Department of Neurology and Centre of Integrated Oncology, University Hospital Bonn, 53127 Bonn, Germany; ulrich.herrlinger@ukbonn.de; 3Department of Neuropathology, University Hospital Bonn, 53127 Bonn, Germany; albert.becker@ukbonn.de; 4Institute of Experimental Oncology, Medical Faculty, University Hospital Bonn, 53127 Bonn, Germany; michael.hoelzel@ukbonn.de; 5Institute of Pathology, Department of Pathology, University Hospital Bonn, 53127 Bonn, Germany; marieta.toma@ukbonn.de

**Keywords:** meningioma, MIB-1, score, recurrence

## Abstract

**Simple Summary:**

Meningiomas are predominantly benign intracranial tumors, and surgical therapy represents the treatment of choice. However, the risk of recurrence and scheduling of follow-up intervals are significantly influenced by immunohistochemical items such as the MIB-1 labeling index. To date, it is not possible to integrate this essential information into the pre- or intraoperative surgical decision making. In the present study, we therefore analyzed baseline variables associated with the MIB-1 labeling index. We found four easily identifiable and routinely recorded risk factors for an increased MIB-1 index and developed a simple and quick-to-use score that allows us to estimate the risk of an elevated MIB-1 index prior to the surgical resection. Furthermore, this score seems to predict the progression-free survival in intracranial meningiomas. We believe that this score might us to more reliably guide patients in preoperative surgical strategy planning and postoperative follow-up scheduling.

**Abstract:**

The MIB-1 index is an essential predictor of progression-free-survival (PFS) in meningioma. To date, the MIB-1 index is not available in preoperative treatment planning. A preoperative score estimating the MIB-1 index in patients with intracranial meningiomas has not been investigated so far. Between 2013 and 2019, 208 patients with tumor morphology data, MIB-1 index data, and plasma fibrinogen and serum C-reactive protein (CRP) data underwent surgery for intracranial WHO grade I and II meningioma. An optimal MIB-1 index cut-off value (≥6/<6) in the prediction of recurrence was determined by ROC curve analysis (AUC: 0.71; 95% CI: 0.55–0.87). A high MIB-1 index (≥6%) was present in 50 cases (24.0%) and was significantly associated with male sex, peritumoral edema, low baseline CRP, and low fibrinogen level in the multivariate analysis. A scoring system (“FORGE”) based on sex, peritumoral edema, preoperative CRP value, and plasma fibrinogen level supports prediction of the MIB-1 index (sensitivity 62%, specificity 79%). The MIB-1 labeling index and the FORGE score are significantly associated with an increased risk of poor PFS time. We suggest a novel score (“FORGE”) to preoperatively estimate the risk of an increased MIB-1 index (≥6%), which might help in surgical decision making and follow-up interval determination and inform future trials investigating inflammatory burden and proliferative activity.

## 1. Introduction

Meningiomas are generally considered to be predominantly benign neoplasms, which account for 36.4% of all central nervous system (CNS) tumors [1,2]. World Health Organization (WHO) grade I and II meningiomas account for 97–99% of all meningiomas and the initial treatment of choice for those meningiomas is gross total microsurgical removal [3,4]. However, benign WHO grade I meningiomas can also recur, with previous investigations reporting a tumor recurrence rate of up to 47% over a 25 year long-term follow-up period [5].

Increased cellular proliferative potential is known to be an important mechanism of oncogenesis [6]. The Molecular Immunology Borstel 1 (MIB-1) labeling index is an established tool for detecting nuclear elements that are exclusively present in proliferating cells. The Ki-67 antigen is present in the nuclei of cells in the G1, S, and G2 phases of the cell division cycle as well as during mitosis. Therefore, the detection of this antigen is a feasible technique in order to determine the growing fraction of a neoplastic cell tissue sample [7,8,9]. Moreover, numerous studies and a recent meta-analysis have shown that the Ki-67/MIB-1 labeling index is an independent predictor of progression-free survival (PFS) in meningiomas [10,11,12,13]. To achieve the best possible long-term outcome with regard to tumor recurrence, appropriate preoperative assessment, effective communication about the aims of surgery, and function-preserving surgery are essential. However, the MIB-1 labeling index is not available in the preoperative period of surgical planning and interactive doctor-patient communication. We therefore evaluated our patient population of sporadic intracranial WHO grade I and II meningiomas with regard to possible preoperative clinical, laboratory inflammatory markers and imaging risk factors for an increased MIB-1 labeling index. Furthermore, we intended to create a proposal for a new score displaying demographic, inflammation, and tumor characteristics in order to identify a population of meningioma patients at increased risk of a high MIB-1 labeling index.

## 2. Materials and Methods

### 2.1. Study Design and Patient Characteristics

Between July 2013 and July 2019, 436 patients were surgically treated for WHO grade I and II meningioma at the neurosurgical department. A review of patient data was retrospectively performed after institutional review board approval had been obtained. The criteria for inclusion in this study were histopathologically confirmed meningioma, intracranial localization, an age greater than 18 years, the availability of the MIB-1 index, preoperative systemic inflammatory parameters (fibrinogen and C-reactive protein), and treatment via a neurosurgical resection. Patients with a neurofibromatosis type 2-associated meningioma and spinal meningiomas were excluded due to differences regarding histopathology and proliferation potential [14,15]. Two hundred and eight patients were included for the data analysis (see Figure 1).

### 2.2. Data Recording

Clinical information including age, sex, comorbidities, Karnofsky performance status (KPS), body mass index (BMI), tumor size, peritumoral edema, tumor growth pattern, WHO grading based on postoperative histopathological examination, immunohistochemical examinations, extent of tumor resection based on the Simpson grading system according to the European Association of Neuro-Oncology (EANO) (Simpson grade 1–3 constitute gross total resection, Simpson grade 4 constitutes subtotal resection, and Simpson grade 5 constitutes biopsy), and postoperative follow-up data were collected and entered into a computerized database (SPSS, version 27 for Mac, IBM Corp., Armonk, NY, USA) [16]. MR imaging was routinely performed within 48 h before surgery. Tumor size was determined using a diameter-based approach in which the single largest diameter on a single axial preoperative contrast-enhanced T1-weighted MR slice was selected [17]. Peritumoral edema was defined as a high signal intensity adjacent to tumors on T2-weighed MRI [18]. Laboratory data collection was performed using the laboratory information system Lauris (version 17.06.21, Swisslab GmbH, Berlin, Germany). Venous blood samples were routinely collected within 24 h prior to the surgical resection of intracranial meningiomas. These laboratory examinations were performed at constant time points, which made it feasible to analyze the probabilities of progression-free survival. The routine examination before surgery included complete blood count, kidney, and liver tests. The coagulation profile (INR, aPTT) was also examined for every subject. The baseline plasma fibrinogen level was determined by the Clauss method, which involves adding a standard and high concentration of thrombin (Dade^®^ thrombin reagent, Siemens Healthineers, Erlangen, Bavaria, Germany) to platelet poor plasma. This fibrinogen concentration was determined based on a reference curve. The serum C-reactive protein values were obtained by turbidimetric immunoassays with a CRPL3 reagent (Roche, Basel, Switzerland) [19].

### 2.3. Histopathology

Histopathological grading was performed based on the 2016 WHO criteria [3]. All pathology reports underwent renewed review to confirm that diagnosis was in keeping with these requirements. Immunohistochemistry was performed in a similar fashion as described before for paraffin-embedded biopsy tissue specimens [20,21]. The MIB-1 labeling index was determined using the following antibody: anti-Ki67 (Clone Ki-67P, dilution 1:1000, DAKO, Glostrup, Denmark). Visualization was conducted with diaminobenzidine, and neuropathological assessment was carried out by an expert neuropathologist (AB). The MIB-1 index was investigated in randomly selected high-power microscopic fields. The proportions of stained and unstained nuclei in the neoplastic cells were determined. The further histopathological workflows were as previously described [22].

### 2.4. Follow-Up

Clinical and imaging follow-up consisted of MRI scans at 3 months after surgery as well as on an annual basis for the following 5 years. Earlier clinical and imaging examinations were advised in case of new or worsened neurological deficits as well as radiological signs of meningioma progression or recurrence. Recurring tumors with radio-clinical correlations, occurring at the site of the initial surgery were considered for analysis. The time to recurrence was defined as the time between the first surgery and the first subsequent treatment (e.g., radiotherapy or re-do surgery). Radiological recurrent tumors without clinical or functional expression, thus not requiring any subsequent therapy, were not included in the analysis [23].

### 2.5. Statistical Analysis

Data were organized and analyzed using SPSS for Mac (version 27.0; IBM Corp, Armonk, NY, USA). Receiver-operating characteristic curves were constructed for the MIB-1 labeling index in the prediction of meningioma recurrence. Cut-off values for the MIB-1 labeling index were set based on the ROC analysis. Normally distributed data are reported as the mean with the standard deviation (SD). Preoperative demographic data, comorbidities, tumor features, and laboratory values were compared between patients with an increased MIB-1 labeling index and those with a normal MIB-1 labeling index using Pearson’s χ2 test (two-sided) for categorical data and independent t-test for continuous data. Receiver-operating characteristic curves were constructed for CRP, fibrinogen, and tumor size. The areas under the ROC curve (AUC) were analyzed, and cut-off values for those variables (CRP, fibrinogen, tumor size) were set based on the ROC analysis. Multivariable binary logistic regression analysis of predictors for an increased MIB-1 labeling index was performed. Dichotomized variables were analyzed using the Wald test. A *p*-value of <0.05 was defined as statistically significant. Significant variables of the multivariate analysis were included in a 5-point score system predicting increased the MIB-1 labeling index. Kaplan–Meier charts of PFS were also calculated. Multivariate Cox regression analysis was performed to analyze the PFS.

## 3. Results

### 3.1. Patient Characteristics

Two hundred and eight patients were surgically treated for intracranial meningioma at our department between July 2013 and July 2019. Median age was 61 years (IQR 51–71), and this study included 152 females (73.1%) and 56 males (26.9%; female/male ratio 2.7:1). The median preoperative Karnofsky performance scale (KPS) at presentation was 90 (IQR 80–100). Further characteristics are summarized in Table 1.

### 3.2. World Health Organization Grades, Tumor Localization, and Extent of Resection

Tumor grading according to the WHO classification criteria included 175 patients with grade I (84.1%) and 33 patients with grade II (15.9%). The area of convexity (31.3%) was the predominant location of intracranial meningiomas in the present study cohort, followed by falx (17.3%) and the sphenoid wing (16.8%). Peritumoral edema was present in 102 (49.0%) patients, and multiple meningiomas were observed in 18 (8.7%) patients. With regard to the extent of resection, Simpson grade I/II resections were performed in 174 patients (83.7%), whereas 34 (16.3%) underwent Simpson grade III/IV resections. Table 1 summarizes the results.

### 3.3. The MIB-1 Labeling Index in the Prediction of Intracranial Recurrent Meningioma

The MIB-1 labeling index was available in 208 patients of the study cohort. The mean (±SD) MIB-1 labeling index was 5.4 ± 2.7%. An ROC curve was constructed, and the AUC of the MIB-1 labeling index in the prediction of tumor recurrence was determined. The AUC of the MIB-1 labeling index ROC curve for intracranial recurrent meningioma was 0.713 (95% CI: 0.55–0.87, *p* = 0.006). Sensitivity and specificity of the MIB-1 labeling index for predicting a recurrent meningioma were 60.0% and 100.0%, respectively (Youden’s index: 0.40), with a threshold of ≥6%. Figure 2A shows the ROC curve and the results of the analysis. Progression-free survival analysis was performed in 182 (91.8%) of the 208 patients. Patients with a MIB-1 labeling index of ≥6% and follow-up data (*n* = 42) had a mean time to progression of 55.7 (95% CI: 43.18–68.17) months, and patients with a MIB-1 labeling index of <6% had a mean PFS time of 72.8 (95% CI: 69.49–76.04) months. Univariate Cox regression analysis showed an increased risk for shorter time to tumor progression in patients with a MIB-1 labeling index of ≥6% (hazard ratio: 5.81, 95% CI: 20.5–16.45, *p* = 0.001). Figure 2B displays the Kaplan–Meier curves of the MIB-1 labeling index groups (<6/≥6%).

### 3.4. Association between the MIB-1 Labeling Index and Clinical, Tumor, and Laboratory Characteristics

In total, 50 (24.0%) patients had a MIB-1 labeling index of ≥6%, and 158 patients had a MIB-1 labeling index of <6%. Patients with an increased MIB-1 labeling index were predominantly male compared to patients with a lower MIB-1 labeling index. Patients with a MIB-1 labeling index of ≥6% had both significantly lower mean baseline serum C-reactive protein (1.6 ± 2.1 vs. 4.3 ± 9.4; *p* = 0.001) and plasma fibrinogen levels (2.5 ± 0.9 vs. 3.2 ± 0.8; *p* < 0.001) compared to patients with a MIB-1 staining index of <6%. Mean (±SD) baseline serum C-reactive protein levels did not differ between male (3.5 ± 8.1) and female (4.00 ± 9.5) patients (*p* = 0.70). Average (±SD) baseline plasma fibrinogen levels of female and male patients were 3.1 ± 0.90 and 2.90 ± 0.87, respectively (*p* = 0.15). Furthermore, univariate analysis showed that patients with a MIB-1 labeling index of ≥6% had significantly larger tumors and presented more frequently with peritumoral edema compared to tumors with a MIB-1 labeling index of <6%. Additional clinical, tumor, and laboratory characteristics in patients with an increased or low MIB-1 labeling index are detailed in Table 2.

ROC curves were constructed, and the AUCs of CRP, fibrinogen, and tumor size in the prediction of an increased MIB-1 labeling index (≥6%) were determined. The AUCs for CRP, fibrinogen, and tumor size were 0.67 (95% CI: 0.59–0.76), 0.73 (95% CI: 0.64–0.82), and 0.65 (95% CI: 0.57–0.74), respectively. The sensitivity and specificity of baseline CRP for predicting a MIB-1 labeling index of ≥6% were 72.0% and 57.0%, respectively (Youden’s index: 0.29). Regarding the baseline fibrinogen for predicting a MIB-1 labeling index of ≥6%, sensitivity and specificity were 68.0% and 73% (Youden’s index: 0.41). The sensitivity and specificity of tumor size for the prediction of a MIB-1 labeling index of ≥6% were 64.0% and 57.6%, respectively (Youden’s index: 0.22). Multivariate binary logistic regression analysis with consideration of sex, KPS (<80/≥80), diabetes mellitus, obesity (BMI ≥ 30.0/<30.0), preoperative corticosteroid medication (yes/no), tumor size (<3.4/≥3.4 cm), brain invasion, fibrinogen (≤2.85 g/L/>2.85 g/L), and CRP (≤1.37 mg/L/>1.37 mg/L) was performed. The multivariate analysis revealed that male sex, peritumoral edema, low CRP (≤1.37 mg/L), and low fibrinogen (≤2.85 g/L) were significantly associated with a MIB-1 labeling index of ≥6%. Table 3 summarizes the results of the multivariate binary logistic regression analysis.

### 3.5. Predictive Score

Further, we created and evaluated a predictive score for an increased MIB-1 labeling index in sporadic intracranial meningioma. The present score was designed with the following intentions: (1) to feasibly predict the MIB-1 labeling index based on easily determinable and routinely acquired preoperative variables and (2) to be easy to calculate in the clinical workflow. These intentions resulted in the following point allocation for a new score, which we called the “FORGE” score, ranging from 0 to 5 points (Figure 3): Preoperative fibrinogen ≤ 2.85 g/L (2 points); preoperative C-reactive protein ≤1.37 mg/L (1 point); male gender (1 point); peritumoral edema (1 point). In the present investigation, the mean score in patients with a MIB-1 labeling index of ≥6% was 3.32 (SD = 1.45), and it was 1.82 (SD = 1.57) in patients with a MIB-1 labeling index of <6% (*p* < 0.001).

The AUC for the FORGE score in the prediction of an increased MIB-1 labeling index (≥6%) was 0.76 (95% CI: 0.66–0.83, *p* < 0.001). Using a cut-off point of 4, the score yields a sensitivity of 62.0%, a specificity of 78.5% (Youden’s index: 0.41), a positive predictive value of 47.7% and a negative predictive value of 86.7%. Figure 4 displays the ROC curve and the results of the analysis. An additive value of <4 implies a probability of 86.7% for not finding a MIB-1 labeling index of ≥6% in the histopathological analysis.

### 3.6. The FORGE Score in the Prediction of Progression-Free Survival

The FORGE score was primarily designed to easily estimate the risk of an increased MIB-1 labeling index. Using a cut-off point of 4, the score yields a sensitivity of 62.0% and specificity of 78.5% with regard to the detection of an increased MIB-1 labeling index. Due to the potential underdetection of high proliferative intracranial meningiomas using a cut-off point of 4, we analyzed the PFS in the study cohort using a dichotomization of the FORGE score into <5 (*n* = 171) vs. 5 (*n* = 18) points. Follow-up MR imaging data were available in 189 (90.9%) of the 208 patients in the study cohort. The mean (±SD) follow-up time was 24.8 ± 21.14 months.

Patients who underwent a Simpson grade I/II resection had a mean time to meningioma progression of 73.67 (95% CI: 69.15–78.20) months, whereas patients who underwent a Simpson grade III/IV resection had a shorter PFS of 51.92 (95% CI: 38.72–65.11) months (univariate Cox regression analysis: hazard ratio = 6.40; 95% CI: 2.31–17.69, *p* < 0.001). Patients presenting with a baseline FORGE score of 5 had mean time to tumor progression of 30.72 (95% CI: 17.51–43.93), and patients with a FORGE score <5 showed a longer mean PFS time of 72.24 (95% CI: 67.84–76.63) (univariate Cox regression analysis: hazard ratio = 8.05; 95% CI: 2.39–27.12, *p* = 0.01). Figure 5 displays the Kaplan–Meier curve of PFS in intracranial meningioma stratified by the FORGE score “0–4 points” and “5 points”.

Multivariate Cox regression analysis of PFS with consideration of age (<65/≥65), KPS (≥80/<80), WHO (I/II), localization (non skull base/skull base), brain invasion (no/yes), dural sinus invasion (no/yes), multiple meningioma (no/yes), the FORGE score (<5/5) and Simpson grade (I and II/III and IV) was performed. A FORGE score of 5 (hazard ratio: 6.75; 95% CI: 1.39–32.73, *p* = 0.02) and a Simpson grade III/IV resection (hazard ratio: 6.47; 95% CI: 1.59–26.29, *p* = 0.009) were significantly associated with shortened time to progression of intracranial meningiomas. Table 4 summarizes the results of the uni- and multivariate analysis.

## 4. Discussion

Negative predictors for tumor progression in meningioma include male sex, young age, low Karnofsky performance status, high WHO grade, high mitotic rate, subtotal resection, and involvement of the optic nerve [24]. A high MIB-1 labeling index is strongly associated with shorter progression-free survival in meningioma and positively correlates with the meningioma grade [13,25,26]. In a typically elective setting, it is particularly important that patients and their relatives are provided with most comprehensive consultation possible. However, the MIB-1 labeling index is not available at the preoperative consultation, and this predictor cannot be considered in the preoperative phase of treatment planning with regard to the extent of resection and potential further radiotherapy. The present study provides a novel scoring system to predict an increased MIB-1 labeling index. This risk index includes the use of four easily identifiable preoperative variables to predict an increased MIB-1 labeling index. Further, this score seems to be capable of predicting the progression-free survival in WHO grade I and II intracranial meningiomas.

Our findings can be summarized as follows: (1) a cut-off value of 6% for the MIB-1 labeling index had the highest specificity and sensitivity to discriminate between nonrecurring and recurring cases; (2) male sex, peritumoral edema, low plasma fibrinogen levels, and low serum C-reactive protein levels were significantly associated with an increased (≥6%) MIB-1 labeling index; (3) the presence of at least 2 variables among male sex, peritumoral edema, and low serum CRP level in combination with a low plasma fibrinogen level was highly predictive of an increased MIB-1 labeling index; (4) the presence of all associated variables (male sex, peritumoral edema, low CRP, low fibrinogen) with an increased MIB-1 labeling index was significantly and independently associated with shorter PFS.

In the present study, we created a ROC curve to determine the appropriate cut-off value of the MIB-1 labeling index in the prediction of meningioma recurrence. The optimal cut-off was set at ≥6% based on this analysis. In the literature, there is an obvious variety of reported MIB-1 labeling index cut-off values in individual studies (2–20%) [13]. A recent meta-analysis summarizing 43 studies and investigating the prognostic value of the MIB-1 labeling index in meningiomas identified a cut-off value set at >4% as appropriate for prognosis prediction with regard to overall survival and progression-free survival [13]. However, other centers also identified an optimum cut-off value at ≥6% [12,27]. The heterogeneity of identified cut-off values in the literature can be also caused by some potential pitfalls associated with specimen sampling and evaluation of the MIB-1 labeling index. In partially or subtotally resected tumors, the sampled neoplastic tissue does not necessarily contain the highest proliferative activity in the area [28]. Furthermore, it should be noted that there is also an interobserver variability regarding the determination of the MIB-1 labeling index due to different techniques (e.g., digital, manual) for its determination [29].

Multivariate binary logistic regression analysis identified male sex, peritumoral edema, low serum C-reactive protein, and low plasma fibrinogen levels as independently and significantly associated with an increased MIB-1 labeling index (≥6%).

This simple association between male sex and high MIB-1 staining indices was also found by some previous studies [30,31]. Kasuya et al. [30] analyzed the growth potential using MIB-1 staining in a consecutive series of 342 meningiomas. They also identified male sex as an independent risk factor for a high MIB-1 labeling index in their logistic regression model.

Furthermore, peritumoral edema was also found to be significantly associated with an elevated MIB-1 labeling index. This finding is also supported by an investigation of Ide et al. [32], in which the association between the edema intensity, tumor size, and MIB-1 was analyzed in 57 histopathologically proven intracranial meningiomas. They found that peritumoral edema is significantly associated with a higher MIB-1 labeling index and the tumor size. In our univariate analysis, tumor size was also significantly associated with higher MIB-1 labeling indices. However, in the multivariate model, only the peritumoral edema was significantly associated with an elevated MIB-1 staining index. This might be explained by the strong association between tumor size and development of a peritumoral edema [32]. To date, the pathogenesis of peritumoral edema in meningioma is still extensively debated and several theories are discussed such as hydrostatic theory, brain compression theory, venous theory, and secretory-excretory theory [33]. Mounting evidence suggests that secretion of vascular endothelial growth factor-A (VEGF-A) by meningioma cells induces angiogenesis and edemagenesis of tumoral as well as peritumoral brain tissue when a cerebral-pial blood supply exists [34,35,36]. A retrospective study of 4 institutions investigating the effect of bevacizumab in atypical and anaplastic meningiomas revealed a decrease in the peritumoral edema on T2-weighted MR-images in 40% of patients [37]. Cytokines such as interleukin-6 can simulate substances such as VEGF-A and MMP-9, which are involved in the pathogenesis of peritumoral edema in meningioma [33,38,39].

In the present study, we found an inverse association of the systemic inflammatory biomarkers serum C-reactive protein and plasma fibrinogen with the MIB-1 labeling index. Patients with an increased MIB-1 staining index (≥6%) had both lower baseline C-reactive protein levels and lower fibrinogen levels. This association might be explained by the secretion of interleukin-6 by human meningioma cells and an autocrine inhibitory regulation of neoplastic cell growth [40]. Using cell culture techniques, it was identified that human meningioma cells are capable of secreting interleukin-6, and the rate of interleukin-6 secretion had an inverse correlation with the growth rate of meningiomas. Furthermore, the addition of an anti-IL-6 antibody enhanced the growth-stimulating effect of meningiomas [40]. Meningiomas are highly vascular lesions and they receive their blood supply from the external carotid artery (middle meningeal artery, accessory meningeal artery, superficial temporal artery, ascending pharyngeal artery, perforating transosseous occipital artery), internal carotid artery (arteries arising from meningohyophyseal trunk, inferolateral trunk, ophthalmic artery), vertebral artery (posterior meningeal artery), or any combination (external carotid -internal carotid artery anastomoses) of these vessels [41]. The extracranial blood supplying arteries do not contain a blood–brain barrier, which make meningiomas permeable to the “periphery” [42]. Moreover, interleukin-6 might also directly influence the integrity of the blood–brain barrier of intracranial blood supplying arteries and might induce changes of the structure and increases the permeability of endothelial cells [43,44]. Those mechanisms make it possible that the interleukin-6 secreted by meningiomas can act on CRP secreting hepatocytes [45]. This pathophysiological mechanism might also be responsible for the inverse association between baseline plasma fibrinogen levels and the MIB-1 labeling index. C-reactive protein and fibrinogen are both linked to the interleukin-6 gene promotor [46]. Recently, it was found that pro-tumor M2 macrophages account for more than 80% of infiltrating tumor-associated macrophages. Furthermore, it was also shown that WHO grade II and recurrent tumor tissues include more M2 macrophages compared to WHO grade I meningiomas and primary tumors. Therefore, the pro-tumoral M2 tumor-associated macrophages play an essential role in tumor growth and recurrence [47]. In contrast, the M1 tumor-associated macrophages may act as tumoricidal cells by promoting inflammation via phagocytosis and cytotoxic cytokine release and recruiting immunostimulating leukocytes to impede growth [48,49,50].

Plasma fibrinogen and serum C-reactive protein are both linked to the interleukin-6 gene promoter, and those parameters may be induced by the autocrine secretion of interleukin-6 by meningioma cells [40]. Moreover, C-reactive protein was found to be capable of polarizing human macrophages to an M1 phenotype and inhibits the transformation to the M2 phenotype [51]. Consequently, those patients in the present study with an elevated CRP and an increased plasma fibrinogen level might have a lower MIB-1 labeling index due to a higher number of M1 phenotype macrophages, which can act as tumoricidal cells via phagocytosis, promoting inflammation, releasing cytotoxic cytokines, and impeding the tumor growth [50,51]. In contrast to the findings with regard to fibrinogen in the present study, Chen et al. [52] investigated the role of preoperative blood tests in predicting the prognosis of atypical meningiomas and found that patients with a higher preoperative fibrinogen level had lower progression-free-survival rate at 3 year follow-up. Therefore, the demonstrated association between fibrinogen and proliferative capability seems to be paradoxical. Interleukin-6, which stimulates the secretion of CRP and fibrinogen, acts as a multifunctional cytokine with stimulatory effects on immune system responses by mediating inflammation and cellular differentiation [33,53,54,55]. An overexpression of interleukin-6 is highly debated as it potentially might act as stimulating growth in approximately 60% of meningiomas, whereas it was also found to be an inhibitor of neoplastic cell proliferation as well [56,57]. However, it has to be reiterated that there is a possible involvement of a confounding bias, since the systemic inflammatory parameters are influenced by several comorbidities and various corticosteroid schedules in different centers.

The created FORGE score in the present study provides a novel scoring system to predict an elevated MIB-1 labeling index and seems to be useful for predicting tumor recurrence in intracranial meningiomas. This score might be useful for preoperative treatment planning and the consultation with patients and their relatives due to the fact that histopathological features are only available in the postoperative period so far. Additionally, patients with an increased FORGE score (≥4) who prefer a watch-and-wait strategy of their incidental meningiomas should be informed about a stringent frequency of the surveillance imaging intervals. The MIB-1 labeling index was also found to be a marker for time to recurrence in a prospective study. This trial investigated the recurrence rates and time to recurrence in WHO grade I-III meningiomas. They identified that patients with a MIB-1 labeling index of 0% to 4%, 5% to 9%, and ≥10% had 2.4, 4.9, and 9.7 recurrences per 100 person-years, respectively. Furthermore, patients with a MIB-1 index ≥5% had significantly more often meningioma recurrences within the first 2 years after surgery compared to patients with a MIB-1 index 0% to 4% [58]. Moreover, a recent retrospective investigation of 239 WHO grade I meningiomas revealed a recurrence rate of 18.8% in patients with a GTR and a MIB-1 labeling index >4.5%, which resulted in a similar risk of recurrence as patients who underwent a subtotal resection. Those findings highlight the need for stringent surveillance of patients with an increased MIB-1 labeling index despite sufficient surgical treatment [59]. Furthermore, adjuvant radiation therapy may be considered in those cases with an increased MIB-1 index or after subtotal resection of meningiomas. Adjuvant radiation therapy after subtotal resection of WHO grade I meningiomas has been found to reduce recurrence [60,61,62,63], although follow-up imaging following subtotal resection still remains the standard treatment in most institutions so far [64]. Against this backdrop, it is of paramount importance to preoperatively inform patients about an increased risk for an elevated MIB-1 labeling index and the subsequent implications with regard to tailored surveillance imaging after surgery and potential need for adjuvant treatment options (e.g., radiotherapy, radiosurgery). Despite new potential techniques such as rapid immunohistochemical methods based on alternating current electric fields in order to intraoperatively determine the MIB-1 labeling index for surgical decision making [65,66], this technique is not established in the intraoperative surgical workflow so far. Therefore, this score might support physicians in preoperative surgical decision making with regard to extent of resection because increased MIB-1 labeling indices were also found to significantly contribute to the emergence of new cranial nerve deficits after Simpson grade I resection of frontal skull base meningiomas [22]. Furthermore, this score might inform future prospective trials investigating the role of inflammatory burden and tumor proliferative activity in meningiomas.

Several limitations existed in our present study. First, although the acquired data were from a highly selective and homogeneous collective, the retrospective design of this study suffered from potential bias due to a single-center experience. Second, other molecular markers such as interleukins, which might give more insight into the role of inflammation and tumor proliferative activity, were not available in this study. Consequently, a multicenter prospective study with a homogeneous design and comprehensive data should further validate the reliability of the FORGE score in patients with intracranial meningiomas.

## 5. Conclusions

The present study found strong correlation between the MIB-1 labeling index and increased risk for shortened progression-free survival in intracranial meningioma patients. Furthermore, we suggest a novel scoring system (“FORGE”), which might enable a guide for preoperative prediction of a high MIB-1 labeling index and thus might improve preoperative patient counseling, surgical decision making, and risk–benefit assessment in the care for intracranial meningioma patients.

## Figures and Tables

**Figure 1 cancers-13-03643-f001:**
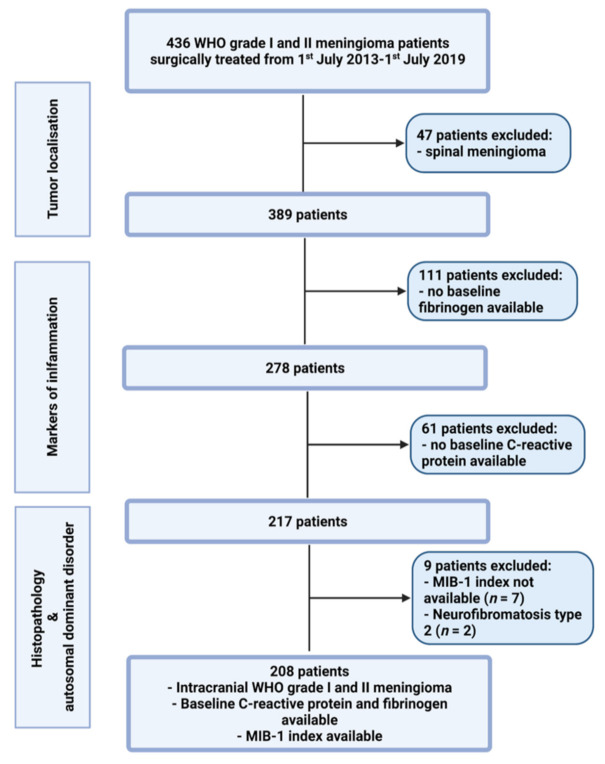
Flow chart illustrating the selection process of consecutive meningioma patients between 1st July 2013 and 1st July 2019.

**Figure 2 cancers-13-03643-f002:**
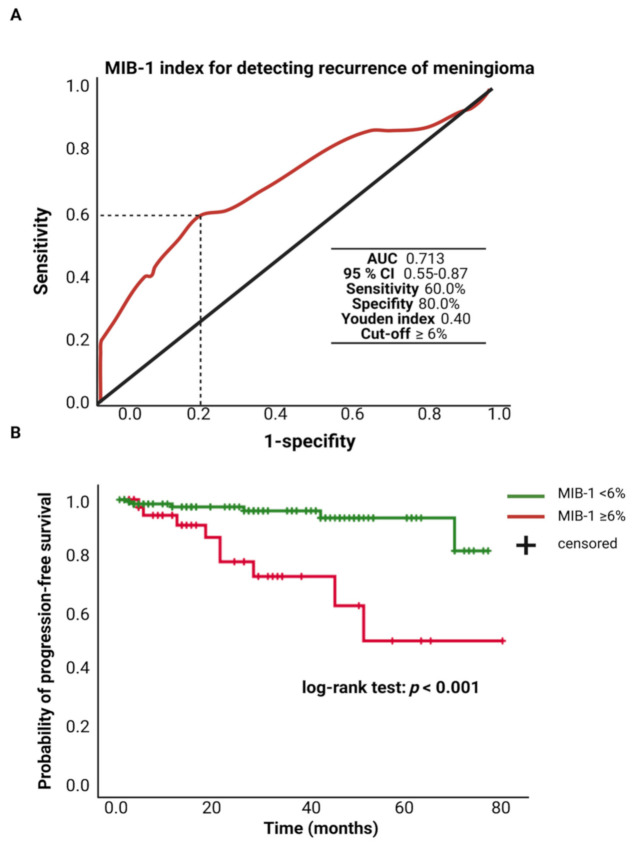
(**A**) Receiver-operating characteristic curve illustrating the MIB-1 labeling index in the prediction of tumor progression of sporadic intracranial meningiomas. (**B**) Kaplan–Meier analysis of tumor progression probability stratified by “MIB-1 ≥ 6%” (red line) and “MIB-1 < 6%” (green line). Vertical dashes indicate censored data (here: progression-free at last follow-up) within the progression-free survival curves. The time axis is right-censored at 80 months. *p* < 0.001 (log-rank test).

**Figure 3 cancers-13-03643-f003:**
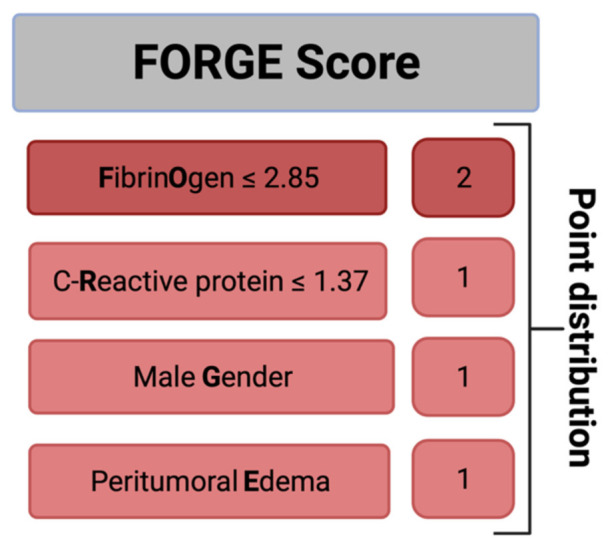
A clinical scoring system to preoperatively estimate the risk of an increased MIB-1 labeling index (≥6%). An additive score value of <4 implies a probability of 87% for not having an increased proliferative potential.

**Figure 4 cancers-13-03643-f004:**
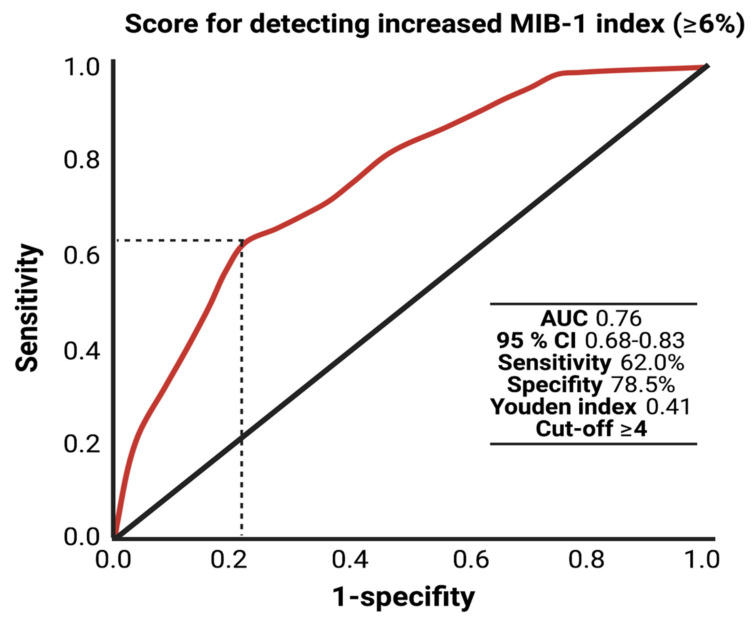
Receiver-operating characteristic curve illustrating the FORGE score in the prediction of an increased MIB-1 labeling index (≥6%).

**Figure 5 cancers-13-03643-f005:**
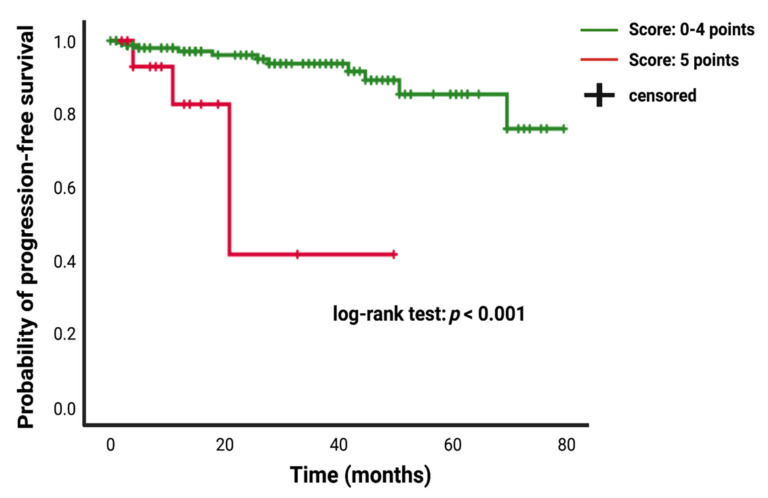
Kaplan–Meier analysis of tumor progression probability stratified by “score: 0–4 points” (green line) and “score: 5 points” (red line). Vertical dashes indicate censored data (here: progression-free at last follow-up) within the progression-free survival curves. The time axis is right-censored at 80 months. *p* < 0.001 (log-rank test).

**Table 1 cancers-13-03643-t001:** Patient characteristics (*n* = 208).

Median Age (IQR) (in y)	61 (51–71)
Sex	
Female	152 (73.1%)
Male	56 (26.9%)
Median preoperative KPS (IQR)	90 (80–100)
Tumor location	
Convexity	65 (31.3%)
Falx	36 (17.3%)
Sphenoid wing	35 (16.8%)
Posterior fossa	29 (13.9%)
Frontobasal	27 (13.0%)
Others	16 (7.7%)
Multiple meningioma	18 (8.7%)
Peritumoral edema	102 (49.0%)
Simpson grade	
Simpson grade I and II	174 (83.7%)
Simpson grade ≥ III	34 (16.3%)
WHO grade	
WHO grade I	175 (84.1%)
WHO grade II	33 (15.9%)

**Table 2 cancers-13-03643-t002:** Baseline clinical, laboratory, and imaging characteristics in patients with an increased and normal MIB-I labeling index. *p*-values in italic and bold represent statistically significant results.

Variable	MIB-I ≥ 6% (*n* = 50)	MIB-I < 6% (*n* = 158)	*p*-Value
Age (mean ± SD)	62.8 ± 16.0	60.3 ± 12.8	0.33
Sex (female/male)	35/23	117/33	***0.02***
KPS (mean ± SD)	86.8 ± 14.2	90.4 ± 11.1	0.10
Diabetes (yes/no)	9/41	16/142	0.21
Smoking (yes/no; available in 204 patients)	12/38	43/111	0.31
BMI (mean ± SD)	26.8 ± 6.1	27.6 ± 6.2	0.41
ASA intake (yes/no)	5/45	21/137	0.63
Dexamethasone (yes/no)	18/32	40/118	0.15
Anticonvulsant drugs (yes/no)	12/38	31/127	0.55
Hemoglobin (mean ± SD)	14.3 ± 1.5	14.0 ± 1.3	0.27
MCV (mean ± SD)	85.8 ± 8.6	87.6 ± 4.9	0.07
Platelet count (mean ± SD)	257.6 ± 78.5	278.2 ± 95.5	0.17
MPV (mean ± SD)	10.6 ± 0.9	12.8 ± 22.0	0.48
Fibrinogen (mean ± SD)	2.5 ± 0.9	3.2 ± 0.8	***<0.001***
C-reactive protein (mean ± SD)	1.6 ± 2.1	4.3 ± 9.4	***0.001***
Tumor size (mean ± SD, mm)	39.8 ± 15.2	32.2 ± 15.1	***0.002***
Peritumoral edema (yes/no)	35/15	67/91	***0.001***
Sinus invasion (yes/no)	13/37	31/127	0.43
Brain invasion (yes/no; available in 206 patients)	6/44	6/150	0.08

**Table 3 cancers-13-03643-t003:** Multivariate binary logistic regression of potential variables predicting an increased MIB-1 labeling index (≥6%). *p*-values in italic and bold represent statistically significant results.

Variable	Adjusted Odds Ratio	95% Confidence Interval	*p*-Value
Sex (male/female)	4.19	1.19–14.82	***0.026***
KPS (<80/≥80)	2.32	0.93–5.81	0.073
Tumor size (<3.4 cm/≥3.4 cm)	1.30	0.48–3.54	0.610
Peritumoral edema (yes/no)	2.90	0.99–8.47	***0.049***
Fibrinogen (≤2.85 g/L/>2.85 g/L)	3.52	1.14–10.90	***0.029***
CRP (≤1.37 mg/I/>1.37 mg/I)	4.90	1.23–19.50	***0.024***
Diabetes (yes/no)	1.29	0.44–3.80	0.649
Brain invasion (yes/no)	1.75	0.36–8.59	0.488
Obesity (BMI ≥ 30.0/<30.0)	1.93	0.55–6.81	0.305
Preoperative corticosteroid medication (yes/no)	1.26	0.43–3.65	0.676

**Table 4 cancers-13-03643-t004:** Uni- and multivariate Cox regression analysis of progression-free survival in intracranial meningiomas. *p*-values in italic and bold represent statistically significant results.

Variable	Univariate			Multivariate		
	Hazard Ratio	95% CI	*p*-Value	Hazard Ratio	95% CI	*p*-Value
**Age (<65/≥65)**	1.32	0.48–3.63	0.59	1.15	0.40–3.35	0.73
**KPS (≥80/<80)**	1.49	0.20–11.37	0.70	1.35	0.17–11.10	0.78
**WHO (I/II)**	1.48	0.47–4.67	0.50	1.77	0.43–7.33	0.43
**Score (<5/5)**	8.05	2.39–27.12	*0.01*	6.75	1.39–32.73	***0.02***
**Simpson Grade (≤II/>II)**	6.40	2.31–17.69	*<0.001*	6.47	1.59–26.29	***0.009***
**Skull base meningioma (no/yes)**	1.68	0.61–4.65		1.26	0.34–4.67	0.73
**Brain invasion (no/yes)**	3.90	0.86–17.71	0.08	2.47	0.42–14.56	0.32
**Dural sinus invasion (no/yes)**	2.65	0.87–8.05	0.09	1.32	0.30–5.93	0.72
**Multiple meningioma (no/yes)**	2.36	0.31–18.24	0.41	1.33	0.16–11.05	0.79

## Data Availability

All data are included in this manuscript.

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
