# Peer review of "FORGE: A Novel Scoring System to Predict the MIB-1 Labeling Index in Intracranial Meningiomas"

_cancers, 2021, doi:10.3390/cancers13143643_

Round 1

Reviewer 1 Report

Study and manuscript are very well contructed. 
A key result of the present study is that a novel score (“FORGE”) might enable to preoperatively estimate the risk of an increased MIB-1 index (≥6%).
The most obvious limitation of the study is that a single-center study without external validation. Please consider external validation.

Overall, I have the impression that this work provides considerable evidence about the MIB-1 labeling index in meningiomas.

Author Response

We agree with the reviewer that the results are limited by a single-center experience. Unfortunately, we cannot provide a validation of the score in an additional cohort with a sufficient number of patients from another time period. However, we have created a study protocol for a prospective controlled trial investigating the FORGE score in primary sporadic cranial meningiomas and we are close to the beginning of the recruitment of patients. Hence, we will be able to validate those findings in a prospective setting which might justify a future clinical application of the FORGE-score in the consultation and care of meningioma patients. We strive to publish those results after the final analysis.

Reviewer 2 Report

I read with much interest the above mentioned manuscript. It a very interesting statistical study, however, shows some weakness. First of all, there are a lot of publications about the concerning subject. In addition the Authors did not mention any pathological mechanism that correlates plasma fibrinogen and serum c-reactive protein with recurrence of meningioma and MIB-1. The Authors did not focus that certain histopathological and anatomical variations lead to a higher probability of recurrence, so they could perform a more complete statistical correlations. Clinical implications of their results, such as neuroradiological follow-up, eventual considerations about post-operative radiosurgery , when subtotal resection is performed, did not take into account. In addition the Authors should better clarified  the concept  of " increased MIB-1 index (≥6%), which might help in the surgical decision making". Really, the complete or partial resection of a meningioma strictly depends on site of lesions and proximity of important or eloquent vascular and nervous structures. In conclusion, the Authors should better define the sentence "Simpson grade resection according to the EANO". 

Author Response

Thank you for reading our manuscript and critically reviewing it, which will help us improve it to a better scientific level and make it more understandable to the reader. We have submitted a revised version of the manuscript and the manuscript underwent an extensive language revision.

In the present investigation we observed an inverse association of the systemic inflammatory markers serum C-reactive protein and plasma fibrinogen with the MIB-1 staining index. Patients with an increased MIB-1 staining index (≥6%) had both lower baseline fibrinogen and lower CRP levels. A potential pathological mechanism might be an autocrine secretion of interleukin-6 by meningioma cells, which results in an autocrine inhibitory regulation of the neoplastic cells (1). Using cell culture techniques, it has been found that human meningioma cells are capable to secrete interleukin-6 and the rate of IL-6 secretion was inversely correlated with the growth rate of meningiomas. Moreover, the addition of an anti-IL-6 antibody significantly enhanced the tumor growth (1). Meningiomas are often highly vascular lesion, which derive their blood supply from extracranial meningeal arteries, which do not contain a blood-barrier, which makes meningiomas permeable to the “periphery” (2). Against this backdrop, this makes it possible that the interleukin-6 secreted by meningiomas can induce the secretion of CRP and fibrinogen. CRP and Fibrinogen are both linked to the interleukin-6 gene promotor (3). High levels of those inflammatory markers might result in a polarization of the macrophages to an M1 phenotype and inhibit the transformation to the M2 phenotype. CRP is known to induce the polarization of macrophages to an M1 phenotype (4). A recent study showed that WHO grade II and recurrent meningiomas include significantly more M2-macrophages. Therefore, we suggest that the meningiomas of those patients with an increased CRP & fibrinogen levels (+ normal MIB-1 index (<6%)) might include significantly more M1-macrophages, which act as tumoricidal cells by phagocytosis, cytotoxic cytokine release and recruitment of immunostimulating leukocytes impeding tumor growth (5-7)

We have also revised the multivariate cox regression analysis of meningioma recurrence and added the following histopathological and anatomical variables to the table 5: presence of brain invasion, dural sinus invasion, multiple meningiomas and location in the skull base. Those parameters were not found to be significant predictors of progression-free survival. Simpson grade III-V resections and a FORGE-Score of 5 remained as the only independent predictors of meningioma progression.

We agree with the reviewer regarding the discussion of potential clinical implications of our results. Therefore, we have revised this part of the manuscript (line: 865-901). Patients with an increased FORGE score (≥4), which was found to be able to estimate elevated MIB-1 staining indices should be informed about a stringent frequency of surveillance imaging intervals in case of an incidental meningioma with a preferred watch-and-wait strategy as the primary treatment approach. Furthermore, a recent retrospective investigation of 239 WHO grade I meningiomas revealed a recurrence rate of 18.8% in patients with an MIB-1 labeling index > 4.5% despite gross total resection was performed. Additionally, they identified that those patients had a similar risk of recurrence as patients who underwent a subtotal resection (8). Adjuvant radiation therapy after subtotal resection of WHO grade I meningiomas has been found to reduce recurrence (9-12). Those findings display the need for stringent follow-up imaging in patients with an increased MIB-1 index or after subtotal resection of meningiomas. Furthermore, adjuvant radiation therapy might be considered for both patients with an increased MIB-1 index or after subtotal resection. Therefore, the FORGE score seems to be a potential additional variable contributing to the preoperative patient consultation with regard to subsequent implications of an increased MIB-1 labeling index (e.g., tailored and stringent surveillance imaging, potential need for adjuvant treatment options).                

Furthermore, we also revised the section regarding MIB-1 index in the surgical decision making (line: 905-909). In a previous institutional retrospective study we have identified that an elevated MBI-1 index significantly contributes to the emergence of new cranial nerve deficits after Simpson grade I resection of frontal skull base meningiomas (13). Therefore, the MIB-1 labeling index might also be considered with regard to the extent of resection and functional outcome.

Additionally, we have revised the definition of the Simpson grading system according to the EANO (line 111-112). Simpson grade 1-3 constitute to gross total resection, Simpson grade 4 constitutes to subtotal resection and Simpson grade 5 constitutes to biopsy.

References:

  1. Todo, T.; Adams, E.F.; Rafferty, B.; Fahlbusch, R.; Dingermann, T.; Werner, H. Secretion of interleukin-6 by human meningioma cells: possible autocrine inhibitory regulation of neoplastic cell growth. J Neurosurg. 1994, 81(3), 394-401
  2. Huang, R.Y.; Bi, W.L.; Griffith, B.; Kaufmann, T.J.; Ia Fougere, C.; Schmidt, N.O.; Tonn, J.C.; Vogelbaum, M.A.; Wen, P.Y.; Aldape, K.; Nassiri, F.; Zadeh, G.; Dunn, I.F.; International Consortium on Meningiomas. Imaging and diagnostics advances for intracranial meningiomas. Neuro Oncol. 2019, 21(Suppl 1), i44-i61
  3. Wong, L.Y.F.; Leung, R.Y.H.; Ong, K.L.; Cheung, B.M.Y. Plasma levels of fibrinogen and C-reactive protein are related to interleukin-6 gene -572C>G polymorphism in subjects with and without hypertension. J Hum Hypertens. 2007, 21(11), 875-82
  4. Devaraj, S.; Jialal, I. C-reactive protein polarizes human macrophages to an M1 phenotype and inhibits transformation to the M2 phenotype. Arterioscler Thromb Vasc Biol. 2011, 31(6), 1397-402
  5. Lisi, L.; Ciotti, G.M.; Braun, D.; Kalinin, S.; Curro, D.; Dello Russo, C.; Coli, A.; Mangiola, A.; Anile, C.; Feinstein, D.L.; Navarra, P. Expression of iNOS, CD163 and ARG-1 taken as M1 and M2 markers of microglial polarization in human glioblastoma and the surrounding normal parenchyma. Neurosci Lett. 2017, 645, 106-112
  6. Ma, J.; Liu, L.; Che, G.; Yu, N.; Dai, F.; You, Z. The M1 form of tumor-associated macrophages in non-small cell lung cancer is positively associated with survival time. BMC Cancer. 2010, 10, 112
  7. Biswas, S.K.; Mantovani, A. Macrophage plasticity and interaction with lymphocyte subsets: cancer as a paradigm. Nat Immunol. 2010, 11(10), 889-896
  8. Haddad, A.F.; Young, J.S.; Kanungo, I.; Sudhir, S.; Chen, J.S.; Raleigh, D.R.; Magill, S.T.; McDermott, M.W.; Aghi, M.K. WHO grade I Meningioma Recurrence: Identifying High Risk Patients Using Histopathological Features and the MIB-1 Index. Front Oncol. 2020, 10, 1522
  9. Ohba, S.; Kobayashi, M.; Horiguchi, T.; Onozuka, S.; Yoshida, K.; Ohira, T.; Kawase, T. Long-term surgical outcome and biological prognostic factors in patients with skull base meningiomas. J Neurosurg. 2011, 114(5), 1278-1287
  10. Soyuer, S.; Chang, E.L.; Selek, U.; Shi, W.; Maor, M.H.; DeMonte, F. Radiotherapy after surgery for benign cerebral meningioma. Radiother Oncol. 2004, 71, 85-90
  11. Park, S.; Cha, Y.J.; Suh, S.H.; Lee, I.J.; Lee, K.S.; Hong, C.K.; Kim, J.W. Risk group-adapted adjuvant radiotherapy for WHO grade I and II skull base meningioma. J Cancer Res Clin Oncol. 2019, 145(5), 1351-1360
  12. Oya, S.; Ikawa, F.; Ichihara, N.; Wanibuchi, M.; Akiyama, Y.; Nakatomi, H.; Mikuni, N.; Narita, Y. Effect of adjuvant radiotherapy after subtotal resection for WHO grade I meningioma: a propensity score matching analysis of the Brain Tumor Registry of Japan. J Neurooncol. 2021, May 17. https://doi.org/10.1007/s11060-021-03775-x
  13. Schneider, M.; Borger, V.; Güresir, A.; Becker, A.; Vatter, H.; Schuss, P.; Güresir, E. High Mib-1-score correlates with new cranial nerve deficits after surgery for frontal skull base meningioma. Neurosurg Rev. 2021, 44(1), 381-387

Round 2

Reviewer 2 Report

I read the revised manuscript and read Authors' response.  Unfortunately, the authors did not well focus on the aspects to be clarified. Specifically, the pathogenesis of the correlation between RCP and FDP and MIB-1. In addition, I noted that they cited old literature (1997) on the subject. They have not inserted or searched for more recent works on the subject to better explain the exact pathogenesis. In the new version of the manuscript there is no new text inserted. I think that it is not suitable for publication. Furthermore, the statement by the authors themselves that the vessels afferent to meningiomas have no blood brain barrier is to be clarified, as there is enormous old and more recent literature which states that cerebral edema caused by meningiomas is due to alteration of the blood brain barrier. "Peritumoral brain edema (PTBE) is a common complication in meningioma and disruption of the tumor-brain barrier in meningioma is crucial for PTBE formation".

"The angiogenic protein vascular endothelial growth factor A (VEGF-A) is believed to be involved in the formation of PTBE around meningiomas, as several studies have found that it is increased in meningiomas with PTBE. VEGF-A is also known as vascular permeability factor due to its ability to increase the permeability of capillaries".

Dan Med J 2013 Apr;60(4):B4626XCV : Intracranial meningiomas, the VEGF-A pathway, and peritumoral brain oedema. Hou J, Kshettry VR, Selman WR, Bambakidis NC. Peritumoral brain edema in intracranial meningiomas: the emergence of vascular endothelial growth factor-directed therapy. Neurosurg Focus. 2013;
35: E2. https://doi.org/10.3171/2013.8.FOCUS13301 PMID: 24289127
2. Yoshioka H, Hama S, Taniguchi E, Sugiyama K, Arita K, Kurisu K. Peritumoral brain edema associated with meningioma.

Author Response

We agree with the reviewer that the literature on the role of interleukin-6 (IL-6) in meningiomas is sparse and therefore partly also old. However, we suggest that there is a pathophysiological link of IL-6 to the proliferation potential due to the fact that CRP and fibrinogen are both linked to the IL-6 gene promotor (1). IL-6 is a multifunctional cytokine with stimulatory effects on immune response system. The influence of IL-6 in meningioma is still unclear. IL-6 has been proven to stimulate growth in human meningioma cell in approximately 60% of meningiomas (2), but on the other hand IL-6 was also found to act as an autocrine inhibitor of tumor cell proliferation in meningiomas (3, 4). Cytokines such as IL-6 can also simulate substances such as vascular endothelial growth factor-A (VEGF-A) and matrix metallopeptidase-9 (MMP-9), which are suggested to be involved in the pathogenesis of peritumoral edema in meningioma (5, 6, 7). Moreover, it has been revealed that IL-6 may directly change the integrity of the blood-brain barrier of intracranial blood supplying arteries and influences their structure by increasing the permeability of endothelial cells (8, 9). Therefore, future investigations on the role of interleukin-6, CRP, and fibrinogen in proliferative activity and edemagenesis of meningioma are needed. Against this backdrop, we are currently planning  a prospective study investigating the FORGE score in primary sporadic cranial meningiomas. Hence, we will be able to provide additional information on the role of inflammation (cytokines, macrophage infiltration, CRP, fibrinogen) on MIB-1 index and peritumoral edema of meningiomas. We strive to publish those results after the final analysis.

Furthermore, we have revised and clarified the part regarding the blood supplying arteries of meningiomas in the lines 366-372. Meningiomas receive their blood supply from the external carotid artery (middle meningeal artery, accessory meningeal artery, superficial temporal artery, ascending pharyngeal artery, perforating transosseous occipital artery), internal carotid artery (arteries arising from meningohyophyseal trunk, inferolateral trunk, ophthalmic artery), vertebral artery (posterior meningeal artery), or any combination (external carotid -internal carotid artery anastomoses) of these vessels (10). Therefore, extracranial blood supplying arteries do not contain a blood–brain barrier, which make meningiomas permeable to the “periphery” (11).  Furthermore, IL-6 might also directly influence the integrity of the blood-brain barrier by inducing changes of the structure and increasing the permeability of endothelial cells (8, 9).

The reviewer is absolutely right that there is enormous literature with regard to pathophysiology of peritumoral brain edema in meningioma and several mechanisms are debated such as hydrostatic theory, brain compression theory, venous theory, and secretory-excretory theory (5). Emerging evidence suggests that secretion of VEGF-A by meningioma cells induces angiogenesis and edemagenesis of tumoral as well as peritumoral brain tissue when a cerebral-pial blood supply exists (12, 13, 14). A retrospective investigation of 4 centers analyzing the effect of bevacizumab in atypical and anaplastic meningiomas revealed a decrease of the peritumoral edema on T2-weighted MR-images in 40% of patients (15). Furthermore, cytokines such as IL-6 can simulate substances such as VEGF-A and MMP-9, which are involved in the pathogenesis of peritumoral edema in meningioma (5, 6, 7).

References

  1. Wong, L.Y.F.; Leung, R.Y.H.; Ong, K.L.; Cheung, B.M.Y. Plasma levels of fibrinogen and C-reactive protein are related to interleukin-6 gene -572C>G polymorphism in subjects with and without hypertension. J Hum Hypertens. 2007, 21(11), 875-82
  2. Boyle-Walsh, E.; Hashim, I.A.; Speirs, V.; Fraser, W.D.; White, M.C. Interleukin-6 (IL-6) production and cell growth of cultured human ameningiomas:-interactions with interleukin-1 beta (IL-1 beta) and interleukin-4 (IL-4) in vitro. Neurosci Lett. 1994, 170(1), 129-32.
  3. Jones, T.H.; Justice, S.K.; Timperley, W.R.; Royds, J.A. Effect of interleukin-1 and dexamethasone on interleukin-6 production and growth in human meningiomas. J Pathol. 1997, 183(4), 460-8.
  4. Todo, T.; Adams, E.F.; Rafferty, B.; Fahlbusch, R.; Dingermann, T.; Werner, H. Secretion of interleukin-6 by human meningioma cells: possible autocrine inhibitory regulation of neoplastic cell growth. J Neurosurg. 1994, 81(3), 394-401
  5. Berhouma, M.; Jacquesson, T.; Jouanneau, E.; Cotton, F. Pathogenesis of peri-tumoral edema in intracranial meningiomas. Neurosurg Rev. 2019, 42(1), 59-71
  6. Bitzer, M.; Wöckel, L.; Luft, A.R.; Wakhloo, A.K.; Petersen, D.; Opitz, H.; Sievert, T.; Ernemann, U.; Voigt, K. The importance of pial blood supply to the development of peritumoral brain edema in meningiomas. J Neurosurg. 1997, 87(3), 368-73
  7. Salpietro, F.M.; Alafaci, C.; Lucerna, S.; Iacopino, D.G.; Todaro, C.; Tomasello, F. Peritumoral edema in meningiomas: microsurgical observations of different brain tumor interfaces related to computed tomography. Neurosurgery. 1994, 35(4), 638-41
  8. Maruo, N.; Morita, I.; Shirao, M.; Murota, S. IL-6 increases endothelial permeability in vitro. Endocrinology. 1992, 131(2), 710-4
  9. Saija, A.; Princi, P.; Lanza, M.; Scalese, M.; Aramnejad, E.; De Sarro, A. Systemic cytokine administration can affect blood-brain barrier permeability in the rat. Life Sci. 1995, 56(10), 775-84
  10. Dubel, G.J.; Ahn, S.H.; Soares, G.M. Contemporary endovascular embolotherapy for meningioma. Semin Intervent Radiol. 2013, 30(3), 263-77
  11. Huang, R.Y.; Bi, W.L.; Griffith, B.; Kaufmann, T.J.; Ia Fougere, C.; Schmidt, N.O.; Tonn, J.C.; Vogelbaum, M.A.; Wen, P.Y.; Aldape, K.; Nassiri, F.; Zadeh, G.; Dunn, I.F.; International Consortium on Meningiomas. Imaging and diagnostics advances for intracranial meningiomas. Neuro Oncol. 2019, 21(Suppl 1), i44-i61
  12. Hou, J.; Kshettry, V.R.; Selman, W.R.; Bambakidis, N.C. Peritumoral brain edema in intracranial meningiomas: the emergence of vascular endothelial growth factor-directed therapy. Neurosurg Focus. 2013, 35(6), E2.
  13. Yoshioka, H.; Hama, S.; Taniguchi, E.; Sugiyama, K.; Arita, K.; Kurisu, K. Peritumoral brain edema associated with meningioma: influence of vascular endothelial growth factor expression and vascular blood supply. Cancer.1999, 85(4), 936-44
  14. Nassehi, D. Intracranial meningiomas, the VEGF-A pathway, and peritumoral brain oedema. Dan Med J. 2013, 60(4), B4626
  15. Nayak, L.;Iwamoto, F.M.; Rudnick, J.D.; Norden, A.D.; Lee, E.Q.; Drappatz, J.; Omuro, A.; Kaley, T.J. Atypical and anaplastic meningiomas treated with bevacizumab. J Neurooncol. 2012, 109(1), 187-93.

Round 3

Reviewer 2 Report

The Authors have now addressed the requested points of weakness.